# Corrosion Inhibition Properties of Corrosion Inhibitors to under-Deposit Corrosion of X65 Steel in CO_2_ Corrosion Conditions

**DOI:** 10.3390/molecules29112611

**Published:** 2024-06-01

**Authors:** Hai Lin, Xiaorong Chen, Zhongming Luo, Jun Xu, Ping Lu, Tianyi Xie, Jiayi Tang, Hu Wang

**Affiliations:** 1CCDC Drilling Engineering Technology Research Institute, Xi’an 710018, China; linhai_gcy@cnpc.com.cn (H.L.); cxr_gcy@cnpc.com.cn (X.C.); xujun_gcy@cnpc.com.cn (J.X.); lup_zcy@cnpc.com.cn (P.L.); 2National Engineering Laboratory for Exploration and Development of Low-Permeability Oil & Gas Fields, Xi’an 710018, China; 3CCDC Geological Exploration & Development Research Institute, Chengdu 610051, China; luozm-sc@cnpc.com.cn; 4Gas Transmission Management Department, Southwest Oil & GasField Company, Petrochina, Chengdu 610215, China; xty@petrochina.com.cn; 5School of Materials Science and Engineering, Southwest Petroleum University, Chengdu 610600, China; tangjiayitjy7@163.com

**Keywords:** under-deposit corrosion, corrosion inhibitor, weight-loss experiment, in situ electrochemical measurements, SEM

## Abstract

Under-deposit corrosion is widely present in the pipelines of oil and gas production, causing significant corrosion damage. In this paper, a novel electrochemical cathodic-polarization method was carried out to accelerate the formation of CaCO_3_ scale on a X65 steel surface in a simulated solution containing scaling ions. Subsequently, pre-scaled X65 steel was placed in a high temperature and pressure autoclave to conduct corrosion weight-loss experiments and in situ electrochemical measurements. The study mainly compared the corrosion inhibition behavior of four quaternary ammonium salt corrosion inhibitors, pyridinium quaternary salt (BPC), quinolinium quaternary salt (BQC), 8-hydroxyquinolinium quaternary salt (BHQ) and pyridinium (1-chloromethyl naphthalene) quaternary salt (1-CPN), in a simulated oilfield scale under corrosive conditions. The results of the weight-loss experiments demonstrated that the inhibition efficiencies of the corrosion inhibitors from high to low were as follows: 1-CPN < BHQ < BQC < BPC. The in situ electrochemical measurements showed that the immersion time and type of corrosion inhibitor had a pronounced influence on the corrosion and corrosion inhibition behavior of X65 steel with CaCO_3_ coating. It was also proved using both EIS and PC that 1-CPN shows the best inhibition performance in all. Lastly, the inhibition mechanism of corrosion inhibitors at under-deposit conditions was analyzed via a surface morphology observation of SEM.

## 1. Introduction

The internal metallic pipelines and related facilities in industrial production often have a tendency to be coated with different types of deposits [1,2,3,4]. These deposits include corrosion products, mineral scales, and products from bacterial growth. Particularly in oil and gas production processes, downhole tubulars, ground processing procedures, long-distance pipeline transportation systems, and associated equipment facilities are prone to developing complicated deposits [5,6,7,8]. The interaction among corrosion, bacteria, and scaling frequently results in complex deposit formations [9,10,11,12,13]. The metal surfaces beneath these deposits are vulnerable to corrosive attack due to the specific occluded environment. This accelerates metal corrosion deterioration.

In order to mitigate internal corrosion attack in oil and gas exploitation, the application of corrosion inhibitors is a common measure [14,15,16,17,18,19,20]. Numerous commercial corrosion inhibitors have been developed and implemented for effective corrosion protection in production. Water-soluble or water-dispersible organic compounds containing N, O, S, or P groups are frequently employed as corrosion inhibitors for metal pipelines. Illustrative examples include imidazoline derivatives, pyridine quaternary ammonium salts, and amines [21,22,23,24,25,26,27,28,29,30]. In practical production processes, the primary agents of corrosion inhibitors are often synergistically combined with small organic molecules such as thiourea and mercaptoethanol to enhance the efficacy while simultaneously reducing costs. Corrosion inhibitors usually protect against corrosion by adsorbing to the surface of the metal to form a protective film, thereby preventing oxygen, water, or other aggressive agents from coming into contact with the metal surface. Despite the attached scale layer, the corrosion inhibitor can still change the electrochemical properties of the metal surface, making it more difficult to be affected by corrosion, thus extending the service life of the metal material. In addition, corrosion inhibitors can also form a protective film on the metal surface with a self-healing or regenerative function that, once damaged, will automatically repair to continue to play a protective role [31,32,33,34,35].

However, in practical production processes, it is frequently observed that corrosion inhibitor formulations exhibiting exceptional performance selected during laboratory assessments fail to meet expectations when implemented on-site. This discrepancy can be attributed primarily to the fact that laboratory evaluations of corrosion inhibitors are predominantly conducted on bare steel surfaces, whereas metal surfaces encountered in practical production processes are typically covered with scales. Consequently, certain conventional corrosion inhibitor products having superior effectiveness may not effectively protect metal from corrosion under such conditions [31,32,33,34,35,36,37,38,39]. Therefore, it is necessary to carry out investigation on evaluation diverse corrosion inhibitor molecules against under-deposit corrosion [40,41,42,43].

K. Chokshi et al. [44] investigated the precipitation process of ferrous carbonate scale and the interaction between corrosion inhibitors and ferrous carbonate deposition, as well as their impact on corrosion rate. The experiments were conducted in an electrolytic cell at 80 °C with a range of ferrous carbonate supersaturation from 7 to 150. A general imidazoline inhibitor was added at different stages to inhibit the formation of iron carbonate scale. The corrosion rate and precipitation rate were measured using electrochemical and weight-loss methods, respectively, and the scales were analyzed using scanning electron microscopy (SEM). The study found that the precipitation rate of iron carbonate was overestimated under the previously used method based on dissolved ferrous ion concentration. Although no antagonistic effect was observed under other test conditions, it can be seen that the addition of corrosion inhibitors delayed the growth of iron carbonate scale. J. E. Wong et al. [45] studied the effect of corrosion inhibitor active components on the growth of iron carbonate scale under CO_2_ conditions. It is debated whether the scale is necessary for the adsorption of the corrosion inhibitor. The author proved through experiments that quaternized amines accelerated the precipitation process of iron carbonate and promoted the existant protective passivation layer. Aiming to study the impact of surface scale deposition caused by supersaturated saline water, as well as the unknown mechanism of its subsequent effect on the corrosion process, M. Ciolkowski et al. [46] proposed a range of laboratory measurements to evaluate the impact of brines on corrosion and scale on both inhibited and uninhibited conditions.

In this paper, the pre-scaling of the CaCO_3_ layer was performed using cathodic polarization on X65 steel. The weight-loss and in situ electrochemical measurements were also carried out to investigate the corrosion behavior and inhibition efficiencies of four quaternary ammonium salts at under-deposit conditions. The top-view and cross-sectional morphologies of X65 steel before and after under-deposit corrosion were observed using SEM at all conditions to analyze the mechanism of corrosion and corrosion inhibition.

## 2. Experimental Methods and Materials

### 2.1. Material Preparation

X65 steel was used in the experiments as the specimen with the chemical composition shown in Table 1. The corrosion experiments were carried out at temperature of 40 °C, total pressure of 1 MPa, CO_2_ partial pressure of 0.5 MPa (total pressure of 1 MPa), and immersion time of 72 h. For the weight-loss experiment, X65 steel samples were prepared at a size of 40 mm × 13 mm × 2 mm and polished gradually using sandpaper ranging from 240# to 1000#. After rinsing with deionized water and degreasing with acetone, they were finally washed with alcohol and dried before being weighed three times. The specimens were sealed using silicone glue with exposed area of 5.2 cm^2^ for accelerated scaling before corrosion experiments in the autoclave (1308550, Dalian Kemao Experimental Equipment Co, Dalian, China). Cylindrical X65 steel samples were used in electrochemical experiments (CS350 in COM3, Wuhan Crest Instruments, Wuhan, China)with exposed area of 1 cm^2^ which underwent the same treatment as in the weight-loss experiment. The solution for pre-scaling treatment consisted of 0.555 g/L CaCl_2_, 0.84 g/L NaHCO_3_ and 34.415 g/L NaCl. The solution for weight-loss experiments in the autoclave consisted of 50 g/L NaCl, 0.406 g/L MgCl_2_·6 H_2_O, 0.852 g/L Na_2_SO_4_, 0.444 g/L CaCl_2_ and 0.0336 g/L NaHCO_3_.

Corrosion inhibitors were synthesized using pyridine, quinoline, 8-hydroxyquinoline, benzyl chloride, and 1-chloromethylnaphthalene. All chemical reagents used were analytical grade reagents provided by Chengdu Kelong Chemical Co., Chengdu, China. All solutions were prepared using deionized water.

### 2.2. Synthesis of Corrosion Inhibitors

By using pyridine, quinoline, and 8-hydroxyquinoline with benzyl chloride in a molar ratio of 1:1 at 110 °C for a duration of four hours, the corrosion inhibitors of (a) pyridinium quaternary salt (BPC), (b) quinolinium quaternary salt (BQC), and (c) 8-hydroxyquinolinium quaternary salt (BHQ) was achieved. Similarly, by subjecting pyridine to a reaction with 1-chloromethyl naphthalene in equimolar proportions at 110 °C for four hours, the formation of pyridinium (1-chloromethyl naphthalene) quaternary salt (d, 1-CPN) was synthesized. The crude product thus obtained by removing the solvent was purified using column chromatographic separation with ethyl acetate and methanol (5:1, *v*/*v*) as eluent. Yields obtained were the following: BPC (67%), BQC (67%), BHQ (67%) and 1-CPN (40%) [47]. The chemical structures of each corrosion inhibitor are shown in Figure 1.

### 2.3. Pre-Scaling

By utilizing the technique of electrochemical cathodic-polarization, CaCO_3_ deposits are generated in situ on the metal surface [48]. Before conducting weight-loss experiments and electrochemical experiments, X65 steel samples were subjected to surface preparation and coated with a CaCO_3_ layer through constant potential polarization on their surfaces for accelerated scaling. Through evaluating corrosion behavior of these samples, it can provide a better simulation for under-deposit corrosion under prolonged operational conditions in production pipelines. In this experiment, a three-electrode cathodic-polarization method was employed to accelerate scaling at scaling durations of 6 h.

The cathodic current was applied to the carbon steel electrode (relative to the saturated calomel electrode at −1.4 V). Ultimately, it induced the reduction in dissolved oxygen. As the pH near the working electrode increased, scale-forming ions in the solution formed deposits on the metal surface as follows:(1)$$O_{2}+H_{2}O+4e\to 4OH^{-}$$

OH^−^ ions were generated near the cathode, causing an increase in pH value. Through the following chemical reaction, solid CaCO_3_ precipitated on the X65 steel electrode:(2)$$Ca^{2+}+HC{O_{3}}^{-}+OH^{-}\to CaCO_{3}(s)↓+H_{2}O$$

After pre-scaling operation, the obtained samples with pre-scaling time of 6 h were carefully rinsed with deionized water to remove any solution and impurities adhering to the surface. After rinsing with alcohol and drying with cold air, the samples were placed in an oven at 35 °C for 2 h, then taken out and placed in a dry vessel for later use. This is necessary to avoid long period exposure of the sample to air.

### 2.4. Weight-Loss Experiments

The weight-loss experiments were conducted under simulated on-site working conditions in a high-temperature and high-pressure autoclave. Three parallel samples were used for the weight-loss experiments, with two of them being calculated for the average corrosion rate after de-film process, and the other sample being washed and dried for characterization of product film. After the aforementioned pre-scaling treatment, three X65 steel samples with scales were suspended in a high-temperature and high-pressure autoclave. Before the corrosion experiment, 1.0 L of corrosive solution was placed into the autoclave (with a total volume of 1.5 L). Then, N_2_ gas with high purity was purged into the solution in the autoclave for 2 h to remove oxygen from the chamber. After that, CO_2_ saturated solution was obtained through another 2 h purging of CO_2_ gas. The heating was then applied to reach the specified temperature, and carbon dioxide was pumped in until it reached a pressure of 0.5 MPa. Pure N_2_ gas was then introduced until the total pressure reached 1 MPa. Timing started when the temperature and pressure inside the autoclave reached the designated conditions.

After soaking for 72 h, the X65 steel samples coated with scales were taken out from the autoclave. Any residual impurities on the surface were rinsed off with deionized water, then rinsed and dried with alcohol. One of the samples was placed in a vacuum oven at 35 °C for 2 h for characterization of the corrosion product film. The other two samples were acid-washed to remove the film, weighed three times, and average corrosion rate was calculated. The formula for calculating corrosion rate (in millimeters per year, mm/a) was as follows:(3)$$CR=\frac{87,600∆w}{\rho St}\;$$
where *CR* was corrosion rate (mm/a), Δw was the weight-loss before and after corrosion experiment (g), *S* represented exposed surface area (cm^2^) and *t* represented the soaking time (h). The inhibition efficiency (*IE*) of corrosion inhibitor can be calculated as follows:(4)$$IE=\frac{{CR}_{0}-{CR}_{inh}}{{CR}_{0}}\;\;\times 100\%\;$$
where ${CR}_{0}$ was the corrosion rate of X65 steel in the solution with absence of corrosion inhibitor (mm/a) and ${CR}_{inh}$ was the corrosion rate of X65 steel in the solution with presence of corrosion inhibitor (mm/a).

### 2.5. Electrochemical Measurements

The in situ electrochemical measurements were conducted in a high-temperature and high-pressure autoclave [10,12]. The electrochemical samples were placed into the working electrode slot, and then sealed and fixed with a fluororubber O-ring (FKM) and a specialized electrochemical pressure cap. The samples coated with a pre-scaling layer were placed in the autoclave with corrosive solution for in situ electrochemical testing in the absence and presence of different corrosion inhibitors. Other operating conditions remained consistent with the weight-loss experiments. The in situ electrochemical measurements used a three-electrode measurement system, with X65 steel coated with scale as the working electrode, platinum wire as the reference electrode, and platinum foil as the auxiliary electrode. All of them are capable of withstanding high temperature and pressure environments, thus achieving in situ electrochemical measurements.

After the temperature and pressure reached the designated values, real-time monitoring of open circuit potential (OCP) began. Linear polarization resistance (LPR) and electrochemical impedance spectroscopy (EIS) tests were performed at certain intervals during the experiment (12 h, 24 h, 48 h, 72 h). LPR measurement was performed by applying a potential ranging from −10 mV to 10 mV (vs. OCP) at a scanning rate of 0.25 mV/s. EIS was carried out at OCP with a frequency range from 10^5^ to 10^−2^ Hz and an AC signal amplitude of ±10 mV. After 72 h soaking, the LPR and EIS measurements were completed, followed by Potentiodynamic Polarization Curve (PC) measurement. The scanning potential range was from −500 mV to +1200 mV (vs. OCP), with a scanning rate of 0.5 mV/s. In the linear polarization region, the corrosion current density can be calculated using the following formula:(5)$$i_{corr}=\frac{B}{R_{p}}$$
where *R*_p_ is polarization resistance (Ω cm^2^) and *B* isthe constant, which can be calculated as follows:(6)$$R_{p}=\frac{∆E}{∆i}$$
(7)$$B=\frac{b_{a}b_{k}}{2.303(b_{a}+b_{k})}$$

*b*_a_ and *b*_k_ are the anodic and cathodic Tafel constants, respectively.

### 2.6. Characterization of Corrosion Products

Scanning electron microscopy (SEM, EVO MA15 ZEISS, Tokyo, Japan) was used to observe the microstructure of the steel surface before and after layer removal. Additionally, the steel samples were sealed with epoxy resin and cut using a cutting machine after the epoxy layer had cured, in order to observe the cross-sectional morphology of corrosion product layers.

## 3. Results and Discussion

### 3.1. Corrosion and Inhibition Behavior by Weight-Loss Measurements

The corrosion rate and inhibition efficiency of X65 steel pre-scaling for 6 h in solutions with different corrosion inhibitors (50 mg/L) are shown in Figure 2. It can be seen that all the inhibitors provide corrosion inhibition for pre-scaled X65 steel, but their effectiveness varies significantly. The order of inhibitory effects for the four inhibitors is as follows: 1-CPN > BHQ > BPC > BQC. Among them, 1-CPN and BHQ have better corrosion inhibition effects, with corrosion inhibition rates reaching 80.22% and 80.15%, respectively. This indicates that the multi-benzene ring structure of 1-CPN plays a protective role in X65 steel after scaling, forming a protective film and producing a shielding effect to inhibit metal corrosion. At the same time, 1-CPN has a strong penetration ability and can effectively reach the metal substrate surface through the scale layer.

### 3.2. Electrochemical Behavior of Corrosion Inhibitors to Scale-Coated X65 Steel

#### 3.2.1. OCP Variations

After 6 h of pre-scaling treatment on X65 steel, in situ electrochemical testing was conducted, and the results of the open circuit potential (OCP) test are shown in Figure 3. It can be clearly observed that during the initial stage of corrosion, the OCP rapidly increases and then starts to stabilize with a slow upward trend around 18 h. The significant fluctuations in OCP during in situ monitoring often indicate major changes in corrosion product films. Therefore, it is evident that corrosion time has a significant impact on steady-state potential. Different protective films with a different protective performance are formed at different stages. Before the electrochemical test, a layer of product film (CaCO_3_) was deposited on the surface of X65 steel through an accelerated scaling process. However, CaCO_3_ does not provide ideal protection to the substrate and may also cause corrosion under the scale, accelerating corrosion behavior. Therefore, significant fluctuations in open circuit potential can be observed during the early stage of OCP testing. The presence of pre-scale (CaCO_3_ film) has a certain influence on the performance of corrosion inhibitors. Some corrosion inhibitors can only adhere to the scale layer, which reduces their effectiveness in inhibiting corrosion. Therefore, the penetrability of corrosion inhibitors through the scale layer is particularly important, as reflected by significant differences in open circuit potential (OCP) under different conditions of quaternary ammonium salts.

#### 3.2.2. Linear Polarization Resistance

LPR tests were conducted at different time periods, and the results are shown in Figure 4. The electrochemical parameters are shown in Table 2. There was a significant difference in *R*_p_ under different corrosion inhibitor conditions, and the results with added inhibitors (50 mg/L) had higher *R*_p_ values than the blank. The relationship between *R*_p_ and corrosion current can be calculated using Formula (5), where *B* is a constant. A larger *R*_p_ value indicates a slower corrosion rate at this time. Therefore, the corrosion inhibition efficiency of different corrosion inhibitors is in the following order from high to low: 1-CPN > BHQ > BQC > BPC, which is consistent with the results of the weight-loss experiments. It is worth noting that the *R*_p_ value gradually decreases over time, indicating a relative weakening of the protective effect of corrosion product film during the 72 h immersion process. This may be because the CaCO_3_ product film generated during the pre-passivation process can provide a certain level of protection to the substrate in the early stages. However, as time goes on, some loosely attached product films formed via partial polarization will detach. At the same time, some corrosion inhibitors cannot penetrate through the scale layer and adhere to the surface product film. These ineffective corrosion inhibitors also detach and enter into the corrosive medium. Therefore, the *R*_p_ value tends to decrease with increasing time.

#### 3.2.3. Electrochemical Impedance Spectroscopy

The in situ electrochemical tests were conducted on X65 steel after pre-scaling for 6 h. The Nyquist and Bode plots of the electrochemical impedance spectra obtained under different corrosion inhibitors (50 mg/L) are shown in Figure 5. The electrical circuit diagram used for fitting the EIS spectra is shown in Figure 6, where *R*_s_ represents solution resistance, *CPE*_dl_ represents double layer capacitance, and *n* is the coefficient. When *n* = 1, *CPE*_dl_ represents pure capacitance with capacitance *Y*^−1^. When *n* = 0, it represents the resistance as *Y*^−1^. When 0 < *n* < 1, it is non-ideal capacitance. When *n* = 0.5, it is Warburg impedance. *R*_ct_ represents charge transfer resistance, *L* stands for inductance, and *R*_f_ represents the film resistance of corrosion products. The fitted electrochemical parameters are presented in Table 3.

As shown in Figure 5, the electrochemical behavior of X65 steel under different corrosion inhibitors exhibits certain differences. However, in the test results, the Nyquist plots obtained under different corrosion inhibitor conditions have very similar shapes, showing a large semicircle in the high-frequency region and an inductive behavior in the low-frequency region [12,13,14]. This indicates that the electrochemical kinetics on the surface of X65 steel are identical. From the Nyquist plot, it can be observed that the semicircle at high frequencies represents the charging and discharging relaxation process on the electrode surface, while the semicircle at low frequencies corresponds to the adsorption of corrosion products on the anode and their coverage on the metal substrate. The appearance of inductance in the low-frequency region is attributed to the presence of a composite layer consisting of ferrous carbonate and calcium carbonate on the metal surface. The low-frequency induction loop indicates the active dissolution of X65 steel substrate in CO_2_ media, with a continuous adsorption of hydrogen ions and other surface substances. Moreover, there is a negative correlation between these surface substances’ coverage degree on the metal surface and potential.

The difference in charge transfer resistance (*R*_ct_) can be visually observed in the Nyquist plot shown in Figure 5. It is generally believed that *R*_ct_ is inversely proportional to the corrosion rate, meaning that a higher *R*_ct_ corresponds to a lower corrosion rate. The fitted values of *R*_ct_ can be seen in Table 3, where the *R*_ct_ value for the blank group is usually around 100 Ω·cm^2^. However, after adding corrosion inhibitors, there is an increase in *R*_ct_. For example, under the condition of using a 1-CPN inhibitor, *R*_ct_ even reaches about six times that of the blank. This indicates that the corrosion inhibitor can penetrate through the pre-existing scale layer and adsorb on the substrate, but there are significant differences in penetration ability among different corrosion inhibitors. For example, under the conditions of BPC corrosion inhibitor, the *R*_ct_ value only increases by less than two times compared to the blank. It is worth noting that under the same conditions, the EIS impedance arc radius decreases with time, which means that the protective effect of the corrosion product film decreases. Generally speaking, it is believed that with increasing time, the protective effect of the corrosion product film will gradually increase. The main reason for this phenomenon in this article is that a CaCO_3_ protective film is formed during the pre-scaling process. It is commonly believed that FeCO_3_ has a better protection effect than CaCO_3_. At the same time, the deposition of CaCO_3_ during the pre-scaling process also has a certain influence on ferrous carbonate sedimentation. Therefore, with the increase in immersion time, some loose CaCO_3_ will fall off, which will also take away some FeCO_3_ attached to the CaCO_3_ layer. The detachment of scale and the inability of newly formed products to effectively deposit on X65 steel matrix result in changes in *R*_p_ and EIS, as mentioned in the article. However, some corrosion inhibitors can still penetrate through the gaps between calcium carbonate scales and act on the matrix. *R*_f_ represents the film resistance value of the product film, and its size reflects the protective nature of the product film. At the same time, with the addition of corrosion inhibitors, *CPE*_dl_ values gradually decrease, indicating a reduction in dielectric constant at the X65 sample corrosion interface. Quaternary ammonium salt molecules have larger sizes and smaller dielectric constants, which can further inhibit corrosion by adsorbing on X65 steel surface.

#### 3.2.4. Polarization Curves

PC measurements were conducted in the final stage of in situ electrochemical testing. The polarization curve test results under different corrosion inhibitors conditions (50 mg/L) are shown in Figure 7. The electrochemical parameters obtained from fitting the PC data are presented in Table 4, indicating differences in PC under different corrosion inhibitor conditions. In the weight-loss experiment, the corrosion rates from smallest to largest were as follows: 1-CPN < BHQ < BQC < BPC < blank. Correspondingly, the corrosion current density (*i*_corr_) values obtained from fitting the polarization curves were 0.018 mA/cm^2^, 0.098 mA/cm^2^, 0.070 mA/cm^2^, 0.323 mA/cm^2^, and 0.531 mA/cm^2^, respectively. The greater the current density, the more severe the tendency for corrosion. Based on OCP analysis, the results indicate that after adding corrosion inhibitors, the relative negative shift in corrosion potential was observed, compared to the blank solution without any additives. Under different conditions of using various types of inhibitors, PC showed consistent trends where increasing potentials resulted in a continuous increase in anodic currents without clear signs of passivation phenomena. The main behavior of anodic processes was active dissolution until reaching higher potentials values. At high potentials, a significant increase in currents was observed due to the rapid deposition of corrosive products on metal surfaces. Literature reports suggest that if |∆*E*_corr_| exceeds 85 mV, it indicates either a cathodic or anodic-type inhibitor. While if |∆*E*_corr_| falls below 85 mV, it suggests a mixed-type inhibitor. These four quaternary ammonium salts belong to mixed-type inhibitors that are primarily cathode inhibitive in nature.

### 3.3. Surface Morphologies of Samples via SEM

#### 3.3.1. Surface Morphologies of X65 Steel Samples after Corrosion Experiments

As for X65 steel samples coated with CaCO_3_ through 6 h of pre-scaling, they were taken into the autoclave for corrosion experiments. The surface morphologies of X65 steel with different corrosion inhibitors for 72 h immersion are shown in Figure 8. It can be observed that different corrosion inhibitors have a significant impact on the surface morphology of the samples. Figure 8a shows the surface of the blank sample after 6 h of scale formation, which exhibits severe corrosion with a large amount of porous and loose corrosion products covering the surface. These are commonly mixed corrosion products consisting of calcium carbonate scale and FeCO_3_ corrosion product scale [49]. After adding the corrosion inhibitor, the surface of the specimen shows varying degrees of corrosion. When BPC is added, the surface of the corrosion product film becomes denser and roughness decreases. However, when BQC and BHQ inhibitors are added, a large amount of flocculent corrosion product film covers the surface, increasing coverage and reducing pore size compared to blank samples. When 1-CPN is added as an inhibitor as shown in Figure 8c, the corrosion product is significantly reduced, with fewer corrosion products and a denser protective film on the substrate.

The surface morphology after removing the corrosion product film of scale-coated X65 steel samples is shown in Figure 9. It can be seen that after film removal, the loose and porous surface corrosion product film on the blank sample without adding any corrosion inhibitor was effectively removed, revealing a clear surface morphology of the substrate after corrosion. There were large corrosive pits and holes all over, indicating uniform corrosion on the metal. The surface of the samples with corrosion inhibitors added after product layer removal changed from flocculent to cellular and exhibited numerous grooves, indicating that the addition of corrosion inhibitors inhibited the corrosion of X65 steel. This resulted in corrosive ions entering the metal substrate surface through scale layers and causing localized corrosion. However, BPC and BQC have limited protective effects on metals as their penetration ability is insufficient to form a uniform protective film. However, the corrosion inhibitors BHQ and 1-CPN exhibit excellent corrosion inhibition performance, with minimal adsorption on the scale surface and almost complete penetration through the scale to reach the metal surface underneath, protecting the metal [50]. After removing the corrosion product film, the sample surface is smooth and even, indicating effective corrosion inhibition, which is consistent with weight-loss and electrochemical results.

#### 3.3.2. Cross-Sectional Morphologies of Scale-Coated Samples after Corrosion Experiments

The cross-sectional morphologies of scale-coated X65 steel samples after being soaked in various corrosion inhibitor solutions for 72 h are shown in Figure 10. It indicates that there exists a layer of corrosion product film on the surface of the corroded samples. In the absence of a corrosion inhibitor, this layer of corrosion product film exhibits a loose attachment to the matrix surface and can be easily cleared off. After adding the corrosion inhibitor, it chelates with the scale layer to form a thicker corrosion product film, which makes it difficult for calcium carbonate scale to detach from the metal substrate. This provides isolation protection and reduces the contact between corrosive media and X65 steel, thereby lowering the corrosion rate. However, different corrosion inhibitors have varying abilities to penetrate the scale layer and provide uniform protection to the substrate. BPC, BQC and BHQ exhibit less than optimal corrosion inhibition effects, resulting in localized corrosion on the surface of X65 steel and an insufficient surface smoothness of the substrate. When using inhibitor 1-CPN, the corrosion product film is dense and evenly distributed on the substrate surface, effectively protecting it and significantly reducing its corrosion rate.

## 4. Conclusions

After 6 h of pre-scaling on X65 steel, the inhibition efficiencies of several corrosion inhibitors on under-deposit corrosion were compared using weight-loss and in situ electrochemical measurements. The conclusions are as follows:(1)Weight-loss experiments have shown that different corrosion inhibitors can effectively inhibit under-deposit corrosion. But there are different capabilities in inhibiting the under-deposit corrosion of X65 steels coated with CaCO_3_ layer. The inhibition efficiencies of the corrosion inhibitors from high to low are as follows: 1-CPN < BHQ < BQC < BPC.(2)The differences in electrochemical behavior under different corrosion inhibitor addition indicate different corrosion inhibition properties to the corrosion under-deposit layer. The pre-scaled CaCO_3_, which forms before corrosion occurs, provides adequate protection in the early stage. However, as immersion was prolonged, a decrease in *R*_p_ and *R*_ct_ occurred, which inferred the increase in corrosion rate. Both EIS and PC results proved the best inhibition efficiency for 1-CPN.(3)The surface morphology of the sample in the absence of a corrosion inhibitor exhibits as a loose and porous layer, which can provide less protection to the substrate. However, in the presence of a different corrosion inhibitor, pre-scaled X65 steels exhibit varying degrees of corrosion. Under the action of corrosion inhibitors, the under-deposit corrosion is inhibited to varying degrees, with BHQ and 1-CPN showing the best inhibition effect to under-deposit corrosion. In the cross-sectional morphologies of BPC and BQC, it exhibited as uneven and the corrosion under the deposit was relatively severe. The corrosion product of 1-CPN covers the substrate surface as a thin and uniform layer. A slightly corroded surface can be observed after the product layer is removed.

## Figures and Tables

**Figure 1 molecules-29-02611-f001:**
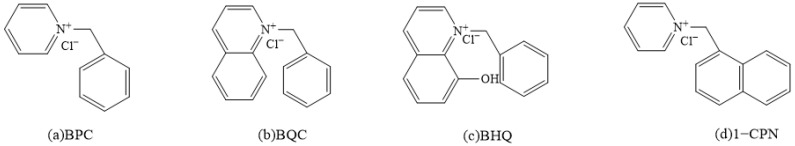
Chemical structures of four synthesized corrosion inhibitors.

**Figure 2 molecules-29-02611-f002:**
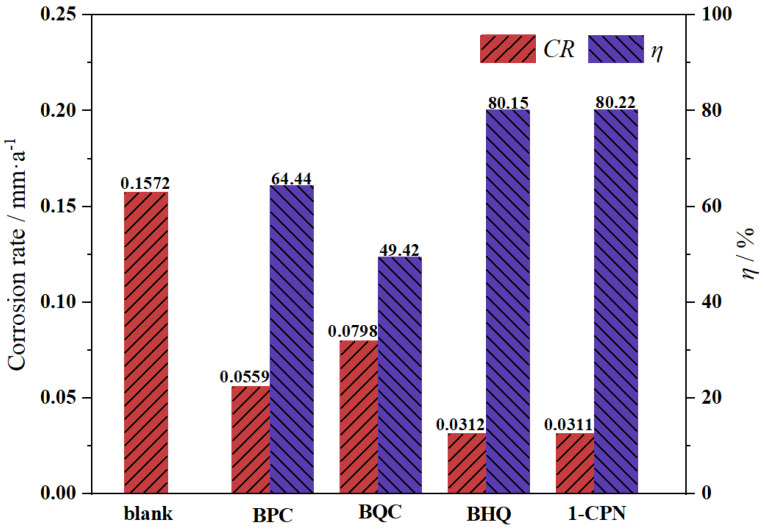
Corrosion rate and inhibition efficiency of pre-scaled X65 steels after immersing in a solution environment at 40 °C, CO_2_ partial pressure of 0.5 MPa (total pressure of 1 MPa) for 72 h, with different corrosion inhibitors (50 mg/L).

**Figure 3 molecules-29-02611-f003:**
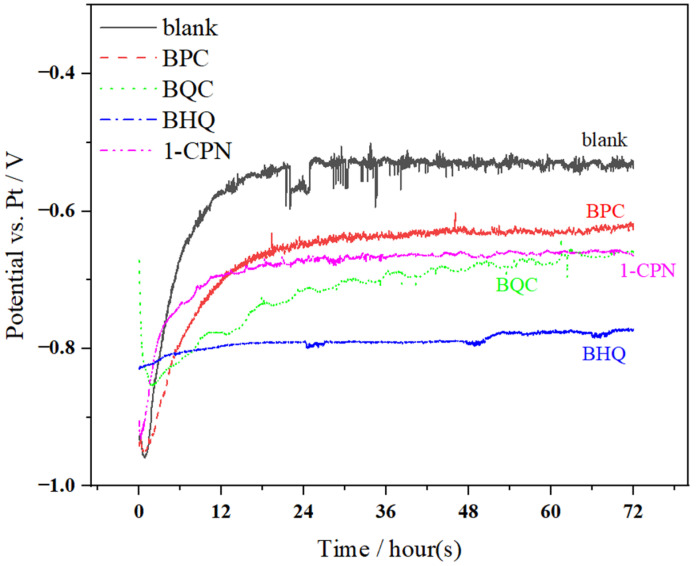
The variation in OCP with time during a 72 h immersion process of X65 steel, pre-scaled for 6 h, at 40 °C and under different corrosion inhibitor conditions (50 mg/L) with a CO_2_ partial pressure of 0.5 MPa.

**Figure 4 molecules-29-02611-f004:**
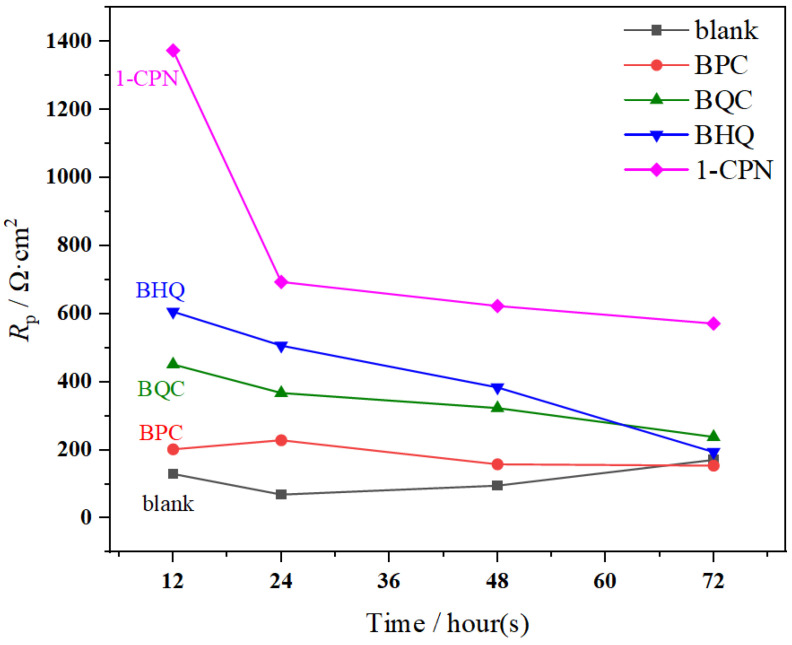
The *R*_p_ values of X65 steel after pre-scaling for 6 h during immersion at 40 °C, with a CO_2_ partial pressure of 0.5 MPa, under various corrosion inhibitor conditions (50 mg/L).

**Figure 5 molecules-29-02611-f005:**
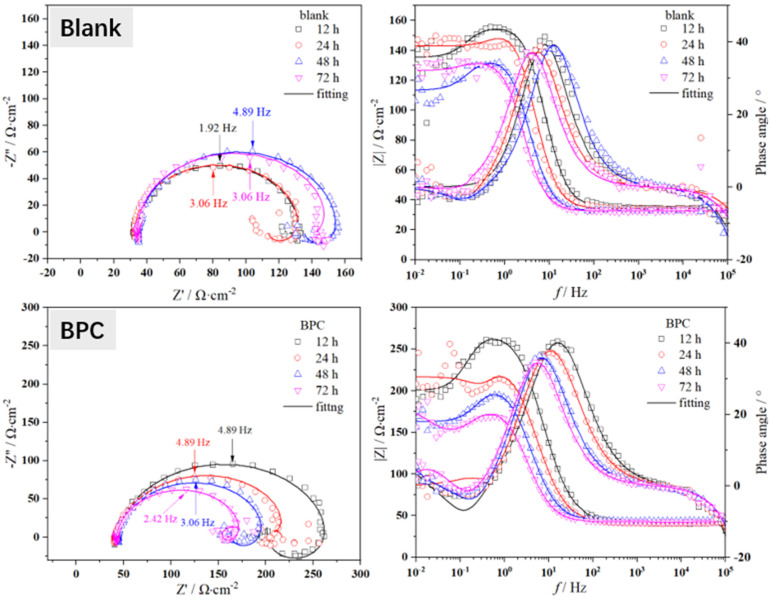
The EIS spectra during a 72 h immersion process of X65 steel, pre-scaled for 6 h, at 40 °C and under a CO_2_ partial pressure of 0.5 MPa with different corrosion inhibitor conditions (50 mg/L).

**Figure 6 molecules-29-02611-f006:**
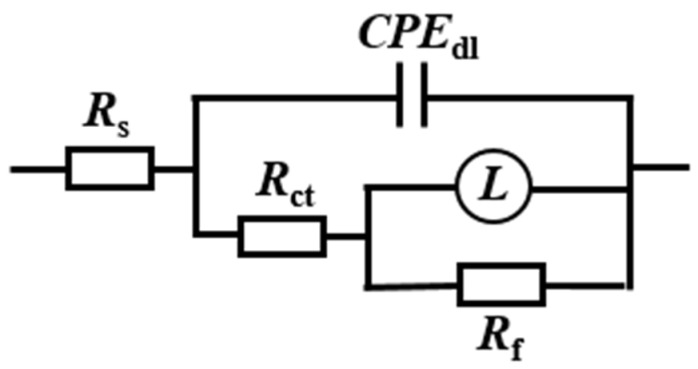
Equivalent circuit diagram.

**Figure 7 molecules-29-02611-f007:**
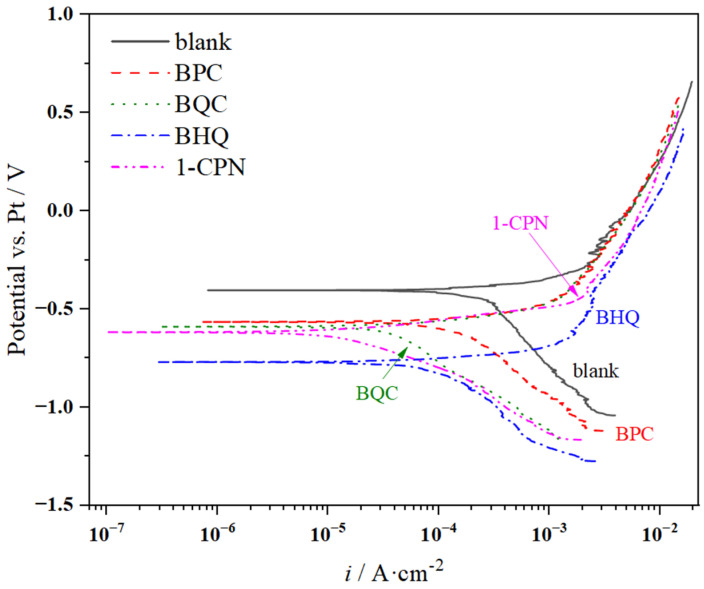
PC comparison of X65 steel under different corrosion inhibitor addition (50 mg/L) after pre-scaling for 6 h.

**Figure 8 molecules-29-02611-f008:**
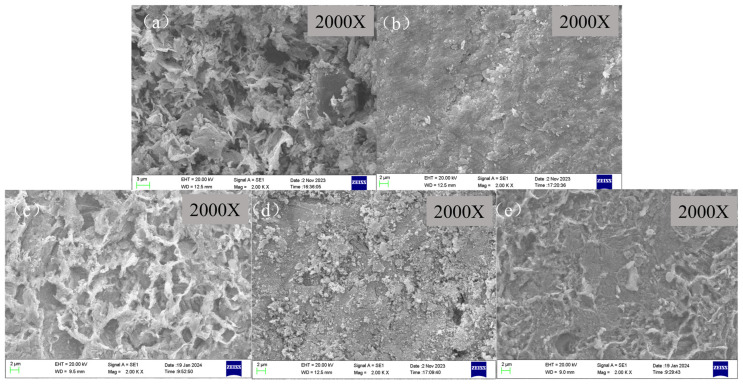
SEM images of the corroded surface before product removal and after immersing scale-coated X65 steel in different corrosion inhibitors (50 mg/L) at a temperature of 40 °C and CO_2_ partial pressure of 0.5 MPa (total pressure of 1 MPa) for 72 h: (**a**) Blank, (**b**) BPC, (**c**) BQC, (**d**) BHQ and (**e**) 1-CPN.

**Figure 9 molecules-29-02611-f009:**
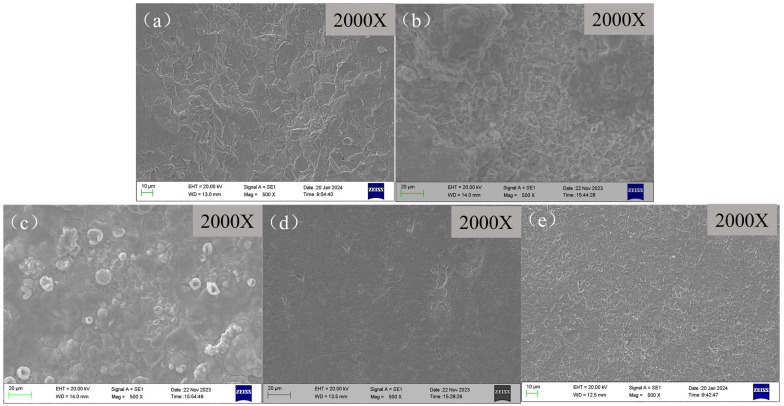
SEM images of the X65 steel surface after removing the layer, which were immersion for 72 h at 40 °C and CO_2_ partial pressure of 0.5 MPa (total pressure of 1 MPa), with different corrosion inhibitors added (50 mg/L): (**a**) Blank, (**b**) BPC, (**c**) BQC, (**d**) BHQ and (**e**) 1-CPN.

**Figure 10 molecules-29-02611-f010:**
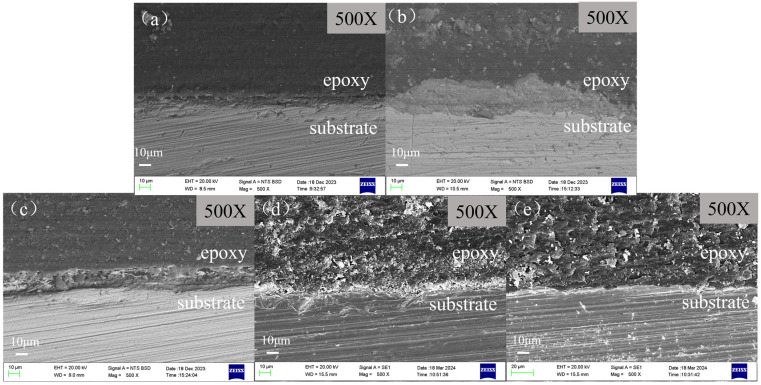
Cross-sectional images of samples before product film removal after immersing scale-coated X65 steel in different corrosion inhibitors (50 mg/L) for 72 h at a temperature of 40 °C and CO_2_ partial pressure of 0.5 MPa (total pressure of 1 MPa): (**a**) Blank, (**b**) BPC, (**c**) BQC, (**d**) BHQ and (**e**) 1-CPN.

**Table 1 molecules-29-02611-t001:** Chemical composition of X65 steel (wt%).

Element	C	Ti	Mn	P	S	Cr	Mo	Ni	N	V	Al	Nb
Content (wt%)	1.60	0.30	0.02	0.004	0.35	0.30	0.25	0.06	0.30	0.025	0.009	0.06

**Table 2 molecules-29-02611-t002:** The electrochemical parameters of the LPR for X65 steel with different corrosion inhibitors (50 mg/L) for samples of 6 h pre-scaling.

Inhibitor	Time (h)	LPR *R*_p_ (Ω·cm^2^)	*IE* (%)
Blank	12	130.32	-
	24	69.855	-
	48	96.203	-
	72	171.55	-
BPC	12	202.73	35.72
	24	229.7	69.59
	48	158.97	39.48
	72	154.78	-
BQC	12	452.11	71.18
	24	368	81.01
	48	323.55	70.27
	72	238.89	28.19
BHQ	12	606.74	78.52
	24	507.2	86.22
	48	384.54	74.98
	72	194.57	11.83
1-CPN	12	1373.6	90.51
	24	694.09	89.94
	48	623.65	84.57
	72	572.08	70.01

**Table 3 molecules-29-02611-t003:** Electrochemical parameters obtained from fitting of EIS spectrum.

Inhibitor	Time (h)	*R*_s_ (Ω·cm^2^)	*CPE*_dl_ (μF·cm^2^)	*n*	*R*_ct_ (Ω·cm^2^)	*L* (H)	*R*_f_ (Ω·cm^2^)	*IE* (%)
Blank	12	35.0	352.0	0.87	102.1	19.2	33.3	-
	24	35.3	414.2	0.94	87.9	10.1	17.3	-
	48	33.2	656.2	0.95	83.1	15.2	20.8	-
	72	32.7	1004.0	0.92	96.1	5.5	14.8	-
BPC	12	41.2	190.7	0.88	163.8	57.0	67.2	33.12
	24	40.7	298.7	0.88	174.4	3.8	23.5	42.72
	48	44.0	418.4	0.92	121.0	21.8	42.6	29.52
	72	42.3	605.1	0.91	120.0	7.1	28.1	20.64
BQC	12	43.7	147.0	0.89	305.0	51.3	76.4	63.06
	24	41.0	204.5	0.91	229.9	33.1	50.5	24.32
	48	45.0	273.2	0.93	267.1	54.6	59.9	31.08
	72	47.0	343.6	0.94	291.1	71.2	49.2	51.20
BHQ	12	41.8	86.4	0.85	554.2	212.2	268.8	77.00
	24	43.0	134.5	0.83	442.4	108.2	192.5	74.62
	48	42.6	262.7	0.84	345.6	80.8	99.2	70.04
	72	42.0	456.9	0.87	122.9	7.3	30.9	21.89
1-CPN	12	46.1	95.0	0.74	1647.0	336.7	319.3	91.90
	24	42.2	165.0	0.77	610.7	14.1	66.4	81.13
	48	47.4	175.7	0.78	552.2	17.0	80.4	80.60
	72	50.4	209.6	0.78	501.4	27.6	115.0	76.66

**Table 4 molecules-29-02611-t004:** The electrochemical parameters of the PC for X65 steel with different corrosion inhibitors (50 mg/L) for samples of 6 h pre-scaling.

Inhibitor	*b*_a_ (mV)	*b*_c_ (mV)	*i*_corr_ (mA·cm^−2^)	*E*_corr_ (V)	*H* (%)
blank	150	1070	0.531	−0.404	-
BPC	211	1562	0.323	−0.566	39.17
BQC	125	921	0.070	−0.590	86.82
BHQ	95	483	0.098	−0.770	81.54
1-CPN	86	269	0.018	−0.618	96.61

## Data Availability

The data presented in this study are available in article.

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
