# Peer review of "Corrosion Inhibition Properties of Corrosion Inhibitors to under-Deposit Corrosion of X65 Steel in CO2 Corrosion Conditions"

_molecules, 2024, doi:10.3390/molecules29112611_

Round 1

Reviewer 1 Report

Comments and Suggestions for Authors

- The authors synthesized four different organic compound without showing the yield percentage, the method of extraction from residulas of reactants. It is recommended to add FTIR, NMR, for proving the product formation

-The authors remove the scale with acid washing without showing the standard or the procedures that they followed.

-the authors avoid long time exposure to air however it is not enough to protect the samples from corrosion especially in the presence of the scale. the authors should put the samples under inert conditions all the time.

- The authors should calculate the corrosion rate from potentiodynamic polarization and it should be in mm/y to avoid unceratinty with LPR.

- the authors should check liturature more and make more discussion about different corrosion inhibitors in underdeposite corrosion.

Comments on the Quality of English Language

- The authors need to clarify the results and put more justifications in the conclusion section.

- There are many spelling and grammatical mistakes need to be revised.

Author Response

Comments 1: The authors synthesized four different organic compound without showing the yield percentage, the method of extraction from residulas of reactants. It is recommended to add FTIR, NMR, for proving the product formation

Response 1: Thank you very much for your advice. The crude product thus obtained by removing the solvent was purified by column chromatographic separation with ethyl acetate and methanol (5:1, v/v) as eluent. Yields obtained: BPC(67%),BQC(67%),BHQ(67%) and 1-CPN(40%). The FTIR of the products is described in the authors' published papers, please refer to the bibliography.

Comments 2: The authors remove the scale with acid washing without showing the standard or the procedures that they followed.

Response 2: Acid descaling is a common chemical descaling method. Its basic principle is that the acid directly with the scale action, and will dissolve the scale. For example: CaCO3 + 2HCl CaCl2 + H2O + CO2.In the paper, a pickling solution (3.5 % urotropin + 10 % HCl in) was used to remove scale and corrosion products from the steel surface.

Comments 3: The authors avoid long time exposure to air however it is not enough to protect the samples from corrosion especially in the presence of the scale. the authors should put the samples under inert conditions all the time.

Response 3: We used vacuum drying at 35°C to preserve the samples. The scale layer is CaCO3, an insoluble substance that does not change easily in acid-free media and does not affect the samples.

Comments 4: The authors should calculate the corrosion rate from potentiodynamic polarization and it should be in mm/y to avoid unceratinty with LPR.

Response 4: Thanks to your suggestion, we have added the effect of LPR on corrosion inhibition efficiency.

Table 2 The electrochemical parameters of the LPR for X65 steel with different corrosion inhibitors (50 mg/L) for samples of 6-hour pre-scaling.

Inhibitor

Time (h)

LPR

Rp(Ω·cm2)

η(%)

Blank

12

130.32

-

24

69.855

-

48

96.203

-

72

171.55

-

BPC

12

202.73

35.72

24

229.7

69.59

48

158.97

39.48

72

154.78

-

BQC

12

452.11

71.18

24

368

81.01

48

323.55

70.27

72

238.89

28.19

BHQ

12

606.74

78.52

24

507.2

86.22

48

384.54

74.98

72

194.57

11.83

1-CPN

12

1373.6

90.51

24

694.09

89.94

48

623.65

84.57

72

572.08

70.01

Comments 5: The authors should check liturature more and make more discussion about different corrosion inhibitors in underdeposite corrosion.

Response 5: We deeply appreciate your suggestion and recognise that a more extensive literature review and discussion of different corrosion inhibitors in the corrosion of composite materials underneath is essential for the improvement of this study. We will seriously consider your suggestions and revise the manuscript to include a more detailed literature review and related discussions to enhance the quality and depth of the study, and hope that you will be satisfied with our efforts and improvements.

Response to Comments on the Quality of English Language

Point 1:The authors need to clarify the results and put more justifications in the conclusion section.

Response 1: After 6 hours of pre-scaling on X65 steel, the inhibition efficiencies of several corrosion inhibitors on under-deposit corrosion were compared by weight loss and in situ electrochemical measurements. The conclusions are as follows:

(1) Weight loss experiments have shown that different corrosion inhibitors can effectively inhibit under-deposit corrosion. But there are different capabilities in inhibiting under-deposit corrosion of X65 steels coated with CaCO3 layer. The inhibition efficiencies of the corrosion inhibitors from high to low are as follows: 1-CPN < BHQ < BQC < BPC.

(2) The differences in electrochemical behavior under different corrosion inhibitor addition indicate different corrosion inhibition properties to the corrosion under deposit layer. The pre-scaled CaCO3, which forms before corrosion occurs, provides adequate protection in the early stage. However, as immersion prolonged, a decrease in Rp and Rct occurred, which inferred the increase of corrosion rate. Both EIS and PC results proved the best inhibition efficiency for 1-CPN.

(3) The surface morphology of sample in the absence of corrosion inhibitor exhibits as loose and porous layer, which can provide less protection to the substrate. However, in the presence of different corrosion inhibitor, pre-scaled X65 steels exhibit varying degrees of corrosion. Under the action of corrosion inhibitors, the under-deposit corrosion is inhibited to varying degrees, with BHQ and 1-CPN showing the best inhibition effect to under-deposit corrosion. In the cross-sectional morphologies of BPC and BQC, it exhibited as uneven and the corrosion under the deposit is relatively severe. The corrosion product of 1-CPN covers on the substrate surface as thin and uniform layer. Slightly corroded surface can be observed after product layer removed.

(Page21, Line 2-21)

Point 2: There are many spelling and grammatical mistakes need to be revised.

Response 1: Thank you very much for your suggestions, we have made changes to the spelling and grammar issues.

Reviewer 2 Report

Comments and Suggestions for Authors

1. In this description, "The corrosion rate and inhibition efficiency of X65 steel pre-scaling for 6 hours in solutions with different corrosion inhibitors (50 mg/L) are shown in Fig. 1." Fig.1—whether to change to Fig. .2?!

 2. Please further explain the role of " synthesized corrosion inhibitors."

3. Table 2. What is "nf"? Please explain what it means.  

4. According to the SEM results, the anti-corrosion effect of Figure 8. (b) is better than that of (C). Looking at the results of Figure 9., we can understand the conclusion of this article, but the observation results of Figure 8 correspond to the conclusion. There are some differences; what are the possible explanations?

Author Response

Comments 1: In this description, "The corrosion rate and inhibition efficiency of X65 steel pre-scaling for 6 hours in solutions with different corrosion inhibitors (50 mg/L) are shown in Fig. 1." Fig.1—whether to change to Fig. .2?!

Response 1: We regret that we have discovered an error in the chart in the text. We will immediately check for errors and make corrections to ensure the accuracy of the chart content. Thank you again for your correction and we will make sure to pay attention to details when revising the manuscript to ensure that the final version is of better quality.

Comments 2: Please further explain the role of " synthesized corrosion inhibitors."

Response 2: We are very sorry that we have not described the role of synthetic corrosion inhibitors in more detail, and have revised our view to say that corrosion inhibitors usually act as an anti-corrosion agent by adsorbing to the surface of the metal to form a protective film, thus preventing oxygen, water or other erosion factors from coming into contact with the metal surface. Despite the attached scale layer, the corrosion inhibitor can still change the electrochemical properties of the metal surface, making it more difficult to be affected by corrosion, thus extending the service life of the metal material. In addition, corrosion inhibitors can also form a protective film on the metal surface, with self-healing or regenerative function, once damaged, will automatically repair to continue to play a protective role. (Page 2, Line 19-26).

Comments 3: Table 2. What is "nf"? Please explain what it means.

Response 3: It should be n in Table 2. We are very sorry that our oversight has caused you problems. n is the coefficient. When n = 1, CPE represents pure capacitance with capacitance Y-1. When n = 0, it represents the resistance as Y-1. When 0﹤n﹤1, it is non-ideal capacitance. When n = 0.5, it is Warburg impedance.

Comments 4: According to the SEM results, the anti-corrosion effect of Figure 8. (b) is better than that of (C). Looking at the results of Figure 9., we can understand the conclusion of this article, but the observation results of Figure 8 correspond to the conclusion. There are some differences; what are the possible explanations?

Response 4: Depicted in Fig. 8 are samples of X65 steel coated with CaCO3 after 6 hours of pre-scaling and placed in an autoclave for corrosion experiments. It does not indicate that the corrosion protection of Fig. 8 (b) is superior to (C), and the final corrosion protection is derived from Fig. 9. Figure 8 shows the effect of corrosion inhibitor on calcium carbonate scale and FeCO3 corrosion product scale two mixed corrosion products, after adding BPC (Figure 8 (b)), the surface of the corrosion product film becomes denser, the roughness is reduced, and the addition of BQC (Figure 8 (c)) inhibitor, a large number of flocculent corrosion product film covered the surface of the specimen, which indicates that the adsorption capacity of corrosion inhibitor BPC in the mixed corrosion product film is more powerful. The obtained SEM images were flatter. However, evaluating the corrosion inhibition performance of corrosion inhibitors on sub-scale corrosion also needs to be illustrated from the aspect of penetration, and we used the surface morphology of the steel sheet after removing the film in Fig. 9 for interpretation.
